# Refractory Hypotension in a Late-Onset Mitochondrial Encephalomyopathy, Lactic Acidosis, and Stroke-like Episodes (MELAS) Male with m.3243 A>G Mutation: A Case Report

**DOI:** 10.3390/brainsci13071080

**Published:** 2023-07-17

**Authors:** Youjie Wang, Enhui Zhang, Chen Ye, Bo Wu

**Affiliations:** 1West China School of Medicine, Sichuan University, Chengdu 610041, China; wangyoujie2002@163.com; 2Department of Neurology, West China Hospital, Sichuan University, Guo Xue Xiang 37, Chengdu 610041, China; 13118305968@163.com (E.Z.); ychenen@foxmail.com (C.Y.); 3Center of Cerebrovascular Diseases, West China Hospital, Sichuan University, Chengdu 610041, China

**Keywords:** MELAS, circulation, hypotension, clinical symptom

## Abstract

(1) Introduction: Symptom spectrum can be of great diversity and heterogeneity in mitochondrial encephalomyopathy, lactic acidosis, and stroke-like episodes (MELAS) patients in clinical practice. Here, we report a case of MELAS presenting asymptomatic refractory hypotension with m.3243 A>G mutation. (2) Case representation: A 51-year-old male patient presented with a headache, vertigo, and difficulty in expression and understanding. The magnetic resonance imaging of the brain revealed an acute stroke-like lesion involving the left temporoparietal lobe. A definitive diagnosis of MELAS was given after the genetic test identified the chrM-3243 A>G mutation. The patient suffered recurrent stroke-like episodes in the 1-year follow-up. Notably, refractory hypotension was observed during hospitalizations, and no significant improvement in blood pressure was found after continuous use of vasopressor drugs and fluid infusion therapy. (3) Conclusions: We report a case of refractory hypotension which was unresponsive to fluid infusion therapy found in a patient with MELAS. Our case suggests that comprehensive management should be paid attention to during treatment. A further study on the pathological mechanism of the multisystem symptoms in MELAS would be beneficial to the treatment of patients.

## 1. Introduction

Mitochondrial encephalomyopathy, lactic acidosis, and stroke-like episodes, known as MELAS, is one of the most common multisystem metabolic diseases caused by mutations in mitochondrial DNA (mtDNA) [1,2]. Pathological mutations of m. 3243 A>G have been identified in nearly 80% of MELAS patients, resulting in impaired integrity of mitochondrial respiratory chain enzyme complex protein and subsequent respiratory chain dysfunction [2]. MELAS syndrome prominently impacts multiple systems that require a high energy supply, notably the brain, heart, and skeletal muscles. This results in distinctive clinical manifestations, including recurring neurological deterioration exemplified by stroke-like episodes and seizure onset, neuropsychological deficits, general skeletal muscle weakness, and cardiomyopathy [1,3,4,5,6,7,8].

In addition to the central nervous system and other frequently affected organs or tissues, the peripheral nervous system [9], as well as the small and large arteries, can be adversely impacted [10,11], suggesting that the clinical symptoms might be of greater diversity and heterogeneity. Although autonomic symptoms can be subtle compared with the central nervous system, autonomic neuropathy has been reported in various mitochondrial disorders concerning both parasympathetic and sympathetic pathways [12]. From a clinical point of view, manifestations, including sexual dysfunction, orthostatic intolerance, diarrhea, bladder incontinence, etc., are recognized as autonomic symptoms [13]. Gastrointestinal symptoms have been reported most commonly in MELAS patients and carriers [14], while there is a scarcity of observations regarding hypotension as a manifestation of autonomic dysfunction in such cases [15]. Accordingly, the presence of episodes of low blood pressure (BP) status in MELAS has not been fully investigated.

Here, we report a late-onset MELAS male case that manifested refractory hypotension. We presented the following case in accordance with the CARE Guideline.

## 2. Case Presentation

A 51-year-old man presented at our hospital with a sudden and persistent headache, accompanied by transient vertigo, difficulty in expression, and understanding for the previous 11 days in March 2021. Meanwhile, he demonstrated personality changes, and irritability, and exhibited symptoms of amnesia, including an inability to recognize his family members. The past medical history was notable for a migraine for 20 years and a progressive bilateral sensorineural hearing loss 5 years prior, for which the patient had been using hearing aids since last year. Ten days before admission to our hospital, an elevated level of blood sugar (fasting plasma glucose: 14 mmol/L) was found at the local hospital. This patient was diagnosed with right parieto-occipital temporal lobe cerebral infarction with acute onset of left central facial paralysis and left limb weakness 2 years ago (Appendix A). Since then, he has been receiving regular antiplatelet, and lipid-lowering treatment and has lived on his own with no obvious sequelae. His wife denied any history of cerebrovascular disease risk factors, such as diabetes, hypertension, dyslipidemia, and atrial fibrillation. The patient’s parents have been deceased for several years, and his mother suffered from epilepsy and migraine during her lifetime, as well as his two sisters.

On admission, this patient was of short stature (Height = 165 cm, Weight = 45 kg, BMI = 16.53 kg/cm^2^), and neurological examination showed sensory aphasia and mild right hemiparesis (V-/V on the Medical Research Council scale 0–V) with right central facial paralysis. The right Chaddock sigh was positive. Advanced and other further neurological examinations were uncooperative because of severe aphasia and irritability. Baseline plasma lactate was 3.9 mmol/L (reference range 0.70–2.10 mmol/L), and fasting plasma glucose was 18.38 mmol/L (reference range 3.90–5.90 mmol/L). The bedside blood pressure (BP) was 113/84 mmHg. The rest of the routine laboratory tests were normal, including liver and renal function, thyroid function, immunologic tests, and serum tumor markers. Transthoracic echocardiography was normal, and 24 h Holter monitoring did not show arrhythmia. Brain magnetic resonance imaging at 3.0 T revealed an acute stroke-like lesion involving the left temporoparietal lobe (Figure 1A), and computed tomography angiography (CTA) showed no obvious evidence of large arterial stenosis and/or occlusion (Figure 1B). Given the medical history, biochemical examination results, and typical imaging findings (not conforming to a specific arterial territory), diagnosis of MELAS (mitochondrial encephalomyopathy, lactic acidosis, and stroke-like episodes) was considered, and further genetic testing was initiated. The muscle biopsy was not performed due to the patient’s poor cooperation and concerns regarding potential risks. The patient’s genetic testing results confirmed the chrM-3243 A>G mutation (Figure 1C) with a peripheral blood heteroplasmy level of 14.98%. Due to the decline in blood heteroplasmy over time, we calculated an age-adjusted blood level of 64.89% [4]. Finally, he was given a diagnosis of MELAS. Treatments of L-arginine (10 g/day) via intravenous infusion, coenzyme-Q10 (30 mg/day) orally, and vitamin B1 (300 mg/day) orally were initiated empirically instantly after the diagnosis of MELAS on day 6.

On the fourth day of hospitalization, this patient experienced sudden acute abdominal pain, with a pale complexion, profuse sweating, painful expression, and yelling. The physical examination showed no significant symptoms of acute abdomen. Nothing remarkable was found with electrocardiogram (ECG), immediate abdominal CT scans, and serum tests (amylase, lipase, C-reactive protein, etc.). The bedside BP examination showed a low level of 87/55 mmHg. A total of 1000 mL of polygeline infusion was administered daily until the fifth day. Additionally, a micro-infusion pump of 200 mg dopamine (diluted in 30 mL of normal saline) per day was used for blood pressure boosting until the 14th day. Bedside ECG monitoring was set to monitor vital signs. In the next few days, the monitoring results showed generally low levels of BP (Figure 2B), with systolic blood pressure (SBP) and diastolic blood pressure (DBP) fluctuating between 90 and 100 mmHg and 60–70 mmHg, respectively, measured with the medical-grade automated BP monitor and recorded by clinical nursing staff at selected time points per day (Figure 2A). There were only mild symptoms of nausea, diarrhea, and abdominal discomfort sensations. After continuous use of vasopressor drugs and fluid infusion therapy for nearly 10 days, as shown in Figure 2, the patient’s BP did not get back to normal physiological level significantly, while there was no obvious symptom of hypotension or might be masked by similar manifestations of stroke-like episodes, such as nausea and vomiting. No significant evidence of body fluids loss, hypovolemia, shock, and sepsis was found. No dizziness, blurred vision, loss of consciousness, and other symptoms were reported. Subsequent echocardiography showed an intact heart structure and normal cardiac function with stroke volume (SV) of 43 mL and ejection fraction (EF) of 58%. Prior medical histories were further questioned. The previous conditions of BP were unknown. No antihypertensive drugs or any treatments that may cause hypotension, such as diuretics, were administered before. His wife denied histories of dizziness and syncope in the patient’s daily life. By reviewing the patient’s history of the last episode 2 years ago, it was found that the overall BP was still at a low level without significant difference from this time, with the lowest of 87/62 mmHg (Appendix A). Consideration was given to refractory hypotension [16]. However, further examinations were not performed due to the family’s financial condition. After comprehensive consideration, the vasopressor drugs were discontinued on day 14. No obvious decreases in BP were detected in the next few days till his discharge on day 16, with neuro-psychological symptoms attenuated and hyperlacticaemia resolved. At the 3-month follow-up, his aphasia and emotional symptoms were almost recovered, and the mRS (modified Ranking Scale) score was 1 point, with a BP level of 99/65 mmHg.

Unfortunately, the patient experienced another stroke-like episode during the one-year follow-up, which took place in February 2022. He presented with symptoms of blurred vision and mild vertigo and was unable to stand or walk firmly due to mild left hemiparesis(V-/IV). Multiple acute infarct-like lesions in the right parieto-occipital lobe and bilateral temporal gyrus were described at the brain MRI (Appendix A). We noticed that his admission BP was 97/69 mmHg. Ambulatory blood pressure monitoring was performed for 24 h this time to gain a more comprehensive understanding of the patient’s blood pressure profile. The mean BP (±SD) was 87 (±5.72)/66 (±5.54) mmHg, ranging from 74/56 to 103/83 mmHg, with the mean arterial pressure (MAP) level of 74 mmHg. No BP load values were found to exceed 135/85 mmHg (daytime) and 120/70 mmHg (night), respectively. No symptoms of dizziness, nausea, vomiting, and loss of consciousness were presented. Echocardiography still showed normal cardiac structure and function (SV 49 mL, EF 63%), and supine-standing blood pressure and electromyography (EMG) test results were negative. In addition, ultrasound bladder scanning found no post-void residual urine. His wife added a history of sexual dysfunction for several years. L-arginine therapy was administered for 5 days along with coenzyme-Q10 and vitamins, and his neurological deficits and symptoms were gradually improved. On the eighth day of hospitalization, he was discharged with a BP level of 88/49 mmHg.

## 3. Discussion

Traditionally known as MELAS syndrome, mitochondrial disease groups with chrM-3243 A>G mutation can present with diverse clinical phenotypes in addition to the typical clinical features we mentioned earlier [17]. Nevertheless, episodes of hypotension remain a relatively rare manifestation. Here, we reported a case of late-onset MELAS presenting refractory hypotension with chrM-3243 A>G mutation. 

Both mitochondrial and nuclear genes mutation can lead to mitochondrial encephalopathy [18]. The ChrM-3243 A>G mutation in the mt-tRNA leucine gene (*MTTL-1*) accounts for about approximately 80% of MELAS cases and could lead to decreased activity of Complex I and Complex IV in the respiratory chain [19]. Stroke-like episodes are the unique feature affecting about 90% of MELAS patients and often serve as the main reason for diagnosis [20,21]. The stroke-like episodes typically occur and recover rapidly without sequela in the early stage of the disease, while leading to progressive neurologic dysfunction and dementia over time [22,23]. The main characteristic of stroke-like episodes is that the brain lesions do not conform to typical vascular territories, which is in accordance with our case [21]. The MRI signals in acute stroke-like episodes involve both cortical and subcortical areas and are dynamic with variable lesions that may even completely disappear during the acute to sub-acute phase [19]. Mitochondrial cytopathy causing neuronal hyperexcitability and cytotoxic damage, and mitochondrial angiopathy causing impaired vasodilation, are believed to be the two main pathological processes associated with stroke-like episodes [24,25,26]. The vascular theory could be explained by the impaired autoregulation caused by endothelial dysfunction of arteries and arterioles [24,27,28]. However, recent research findings that neuronal deficit is more prominent than angiopathic changes in MELAS suggest that the paroxysmal stroke-like episodes may primarily attribute to neuronal necrosis [29].

Due to the impact of the mitochondrial disorder on different tissues, MELAS patients present a variety of clinical manifestations and frequently more than one primary symptom simultaneously. Alongside the typical features of lactic acidemia, ragged red fibers in the biopsy, and recurrent stroke-like episodes, previous studies demonstrated other frequent manifestations, including dementia, seizure, disturbance of consciousness, cognitive impairment, and exercise intolerance [17,30,31]. In our case, the clinical presentations of hearing loss, headache, cognitive impairment, and neuropsychiatric symptoms such as aggressiveness and irritability were also suggestive of MELAS syndrome [17]. Generally, our patients exhibited representative symptoms induced by mitochondrial-associated metabolic disorders.

To the best of our knowledge, hypotension attributed to a component of autonomic dysfunction in MELAS patients was first reported by Zelnik et al. [15]. Subsequently, hypotension was occasionally reported, with these case studies providing more specific underlying causes for hypotension (for example, hypothyroidism [32] and sepsis [33]), and the blood pressure abnormalities or variations were not discussed in-depth. On the contrary, adult MELAS patients have been reported with elevated rates of hypertension [34]. Compared with other types of mitochondrial encephalopathy, the prevalence of hypertension in MELAS patients with m.3243 A>G mutation is remarkably higher [35]. The increased risk of hypertension is believed to be on account of mitochondrial angiopathy. In MELAS syndrome, reduced plasma concentrations of citrulline consequently can influence the synthesis of L-arginine via the urea cycle, which acts as a nitric oxide (NO) precursor, and its deficiency further lead to endothelial damage [36]. In addition, excess cytochrome c oxidase (COX) activity accompanied by mitochondrial proliferation in endothelial and smooth muscle cells can result in increased NO binding of NO and a relative lack of its availability [37]. The insufficiency of NO exerting a vasomotor regulatory effect subsequently increases the risk of hypertension. While in our case, episodes of hypotension were detected in the patient during all three hospitalizations, and there was no significant improvement observed with treatment. As illustrated in Figure 2, the patient’s episodes of hypotension during the second hospitalization were most pronounced around the 5th and 14th days. We propose possible mechanisms to explain the puzzling refractory hypotension of our patient as follows and provide a comprehensive summary in Figure 3.

The patient in our case was short in stature and emaciated, which is consistent with the common characteristics identified in most MELAS patients with m.3243A>G mutation [31]. In addition, this patient experienced inadequate nutrition intake and prolonged bedridden periods due to recurrent stroke-like episodes. We initially considered the reduction of effective circulating blood volume (ECBV) as the major cause of hypotension. However, there was no significant improvement in blood pressure, despite administering fluid infusion and vasopressor therapy. Additionally, the ECG showed no functional or structural abnormalities of the heart.

Since mitochondrial angiopathy is recognized to increase the risk of hypertension, it is rational to postulate that the refractory hypotension in our patient might be caused by another leading pathophysiology in MELAS, namely the neuronal hyperexcitability, and cytopathy. Therefore, we propose a hypothesis that the refractory hypotension in our patient may be related to the autonomic manifestations in MELAS. Because of the vulnerability of neurons to hypoxia, neurological deficits resulting from mitochondrial disorder are commonly observed in MELAS [30]. Typical major central nervous system manifestations of MELAS include seizures, recurrent headaches, altered consciousness, and cognitive impairment [17]. Autonomic nervous system symptoms are also reported at a higher rate in MELAS patients, including orthostatic dizziness, abdominal pain and/or cramps, diarrhea, bladder incontinence, male sexual dysfunction, and pupillary dysfunction [14,38]. In spite of the less frequency of studies reporting hypotension as a manifestation of autonomic dysfunction in MELAS patients, the existence of chronic hypotension has been found to be related to autonomic dysregulation caused by distributed autonomic pathways and/or loss of peripheral noradrenergic fibers [39]. This was the case for our patient: The clinical presentations of sexual dysfunction, diarrhea, and abdominal discomfort sensations have been described as autonomic symptoms in patients with m.3243 A>G mutation [14]. Additionally, gastrointestinal symptoms were speculated to be related to neurological deficits [40]. Therefore, we supposed that nerve cell necrosis, particularly damage in central autonomic pathways as the norepinephrine level was normal in the patient [41], involves the contractile dysfunction of vascular smooth muscle cells and further causes a decrease in blood pressure. It is supported that the exacerbations of hypotension coincide with the onset and recovery of stroke-like episodes, which could reflect a severe level of cytopathy according to the neuronal cytopathy theory.

Medication is also a potential contributing factor to hypotension. The use of arginine, a NO precursor in vivo, has been recommended for managing stroke-like episodes in MELAS patients [21,42]. The NO-mediated biological effects can still be increased with supplementation of exogenous L-arginine despite the fact that nitric oxide synthase (NOS) is saturated theoretically, which is the phenomenon named “L-arginine paradox” [43]. It has been reported that both oral and intravenous arginine could improve stroke-like episodes symptoms [44,45]. We suppose that the vessel relaxation effect of NO as a consequence of intravenous L-arginine therapy in our patient might also contribute to the episodes of hypotension partly. It is important to note that the potential adverse effects of L-arginine that have been previously reported include hyperkalemia, metabolic acidosis, and even sudden death [46]. Other medication factors in our case, such as routine antiplatelet and lipid-lowering therapy conducted between the two stroke-like episodes, did not have a significant impact on the blood pressure. Whether arginine had an effect on the decreased blood pressure in MELAS patients remains unclear.

We would like to acknowledge the limitations of our study. Due to the blood pressure measurements being taken at selected time points per day in our case rather than dynamic monitoring, it is still not possible to completely exclude the possibility of hypotension related to intra-day and visit-to-visit blood pressure variability [47]. Moreover, while the manifestations concerning the autonomic system partly support the hypothesis of autonomic dysfunction, our case lacks other relevant examinations to negate other potential factors of hypotension, including measuring adrenocorticotrophic hormone (ACTH) levels to rule out Addison’s disease and assessing capillary refill time to thoroughly exclude hypovolemia [48,49,50].

## 4. Conclusions

In conclusion, it seems possible that the rare episodes of hypotension in this MELAS patient were due to the vasoconstrictive dysfunction effect caused by nerve cell cytopathy, and it may overwhelm the blood pressure-raising effect of mitochondrial angiopathy. Other possible comorbidities affecting blood pressure cannot be excluded. In response to the multisystem symptoms in MELAS, we propose that comprehensive management should be paid attention to during treatment, and drugs that may interfere with the respiratory chain and increase mitochondrial burden should be generally avoided. Currently, there are still many unanswered questions about the specific mechanisms behind the varied clinical manifestations of MELAS. A further study with more focus on the pathological process is therefore recommended for better carrying out treatment and improving prognosis.

## Figures and Tables

**Figure 1 brainsci-13-01080-f001:**
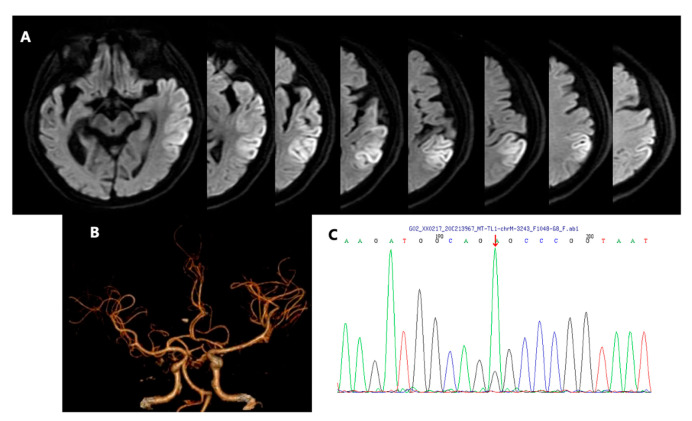
Brain diffusion-weighted image (DWI) of left temporoparietal lobe infarction lesion (**A**). Computed tomography angiography (CTA) showed no obvious evidence of large arterial stenosis and/or occlusion (**B**). Subsequent genetic testing results confirmed the chrM-3243 A>G mutation. The red arrow indicates the position of mutation (**C**).

**Figure 2 brainsci-13-01080-f002:**
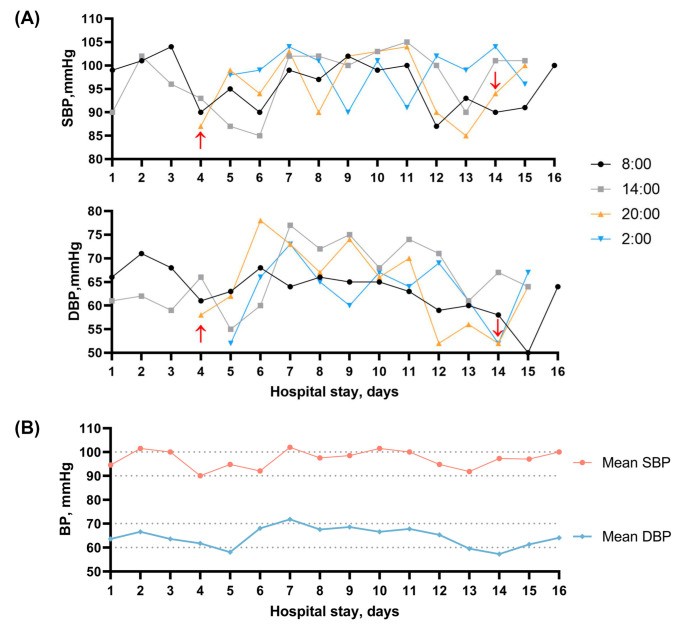
(**A**) Changes in blood pressure levels after admission. Red arrow: start and end of continuous use of vasopressor drugs and fluid infusion therapy; (**B**) Mean blood pressure measurements taken at different time points each day during the hospitalization.

**Figure 3 brainsci-13-01080-f003:**
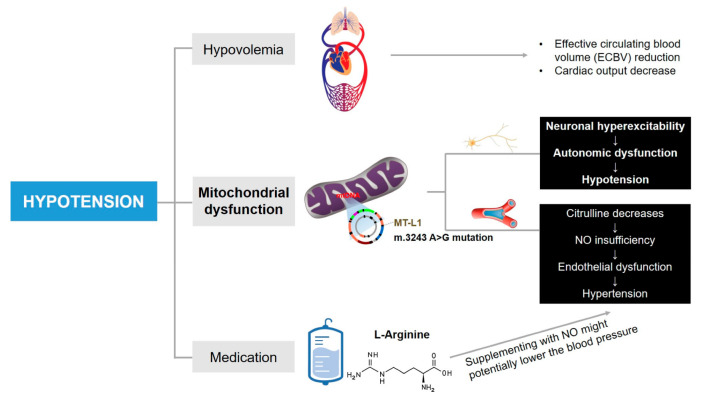
The mechanisms potentially lead to episodes of hypotension in this case. Other underlying factors cannot be ruled out.

## Data Availability

All data related to this case report are documented within this manuscript.

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
