# Peer review of "Refractory Hypotension in a Late-Onset Mitochondrial Encephalomyopathy, Lactic Acidosis, and Stroke-like Episodes (MELAS) Male with m.3243 A>G Mutation: A Case Report"

_brainsci, 2023, doi:10.3390/brainsci13071080_

Round 1
Reviewer 1 Report
In this case study, the authors report on a patient with MELAS who presents with asymptomatic refractory hypotension.
My main concern is that the authors are claiming (to their knowledge) that this is the first reported case of refractory hypotension in MELAS. Hypotension in MELAS has been previously reported by: Ge YX et al. 10.1016/j.ensci.2016.11.005 and also Goel H et al. 10.4158/EP171955. Could the authors adjust their manuscript to reflect this information.
Could the authors also address the following please:
· Line 82. Presumably the heteroplasmy level of 15% was determined in blood leukocytes, rather than serum.
· Line 83. Please clarify whether the treatments were administered orally or intravenously.
· Figure 2. Could the authors discuss why the blood pressure recordings are presented as multiple lines.
· Line 80. Please rephrase this sentence to state that the patient did not wish to undergo a muscle biopsy.
The English requires moderate editing.
Reviewer 2 Report
I have the following comments:
- Please check relevance of references included (i.e. number 3; 12) and make sure all the references are reported in the right place
- In "introduction" I would emphasize more what is known from literature about the autonomic involvement in 3243 mutation and its pathophysiology
- Line 47: what do you mean with "stable low blood pressure"? Please clarify in the text.
- Line 80-84: Why did you want to perform a muscle biopsy? It's not necessary for the diagnosis as other tissues can be used (i.e. blood, urine). Please calculate the age corrected heteroplasty in blood (see PMID 29735722, Grady JP et al 2018) as this is a better indicator of disease severity. Have you measured the heteroplasmy in other tissues (i.e. urine)?
- Clarify if/when patient was symptomatic regarding the low blood pressure during the admission.
- Were other causes of hypotension excluded? i.e. Addison's, infections... If not add this to limitations. What was the BMI of your patient? Could the reduced blood pressure be secondary to low BMI/malnutrition?
-Line 219: there is a phrase starting with "And"
- It would be nice to have a figure summarizing the different hypothesis you discussed that could explain the low blood pressure in your patient.
- At the end of discussion a limitation paragraph should be added
The article needs major editing of English language.
Reviewer 3 Report
The case report by Wang et al. Analizes the occurrence of refractory hypotension in a patient with MELAS. I have some comments to be addressed to improve the manuscript.
- Line 53. Please refer to it as “amnesia”, and then specify: “not knowing his family members”.
- Line 69-70. Did authors measure the capillary refill time at this stage?
- Line 94. Please report the dosage of dopamine and the amount of colloids administered. Please specify in which way the episodes of hypotension were “refractory”.
- Line 97. A BP of 105/74 doesn’t seem to be low. Please only report the episodes of real hypotension and the duration of those episodes.
- Line 110-111. The same as above.
- Line 178. Please replace “the hypotension” with “episodes of hypotension”.
- In this case, Assessment of Capillary Refill Time would have been an easy tool (doi: 10.1186/s12871-022-01920-1) to investigate periferal perfusion during episodes of hypotension (doi: 10.1136/emj.2007.055244 - doi: 10.1016/j.ajem.2007.06.026). Please discuss it as a limitation and add these 3 references.
Round 2
Reviewer 1 Report
The authors have addressed my feedback
Author Response
Thank you for your confirmation that we have adequately addressed your feedback. We appreciate the time and effort you took in reviewing our manuscript and providing valuable comments.
As the journal's system still prompts for a response, we would like to reiterate our gratitude for your contribution to the improvement of our work. We welcome any other comments, as we have made some minor modifications in response to the second round of comments from Reviewer 2.
Reviewer 2 Report
I am happy with the revisions. I have some comments:
Line 36-41: please rewrite the phrase, it's not clear
Line 66: Introduce the phrase with something like "The past medical history was notable for..."
Line 82: report normal values of your laboratory
Line 145-146: please rewrite the phrase, it's not clear
Line 149: change with something like "described at the brain MRI"
Line 264: change frequent with frequency
Overall, I find the discussion too long. There is a long description of the pathophysiology of stroke-like episodes that it's beyond the scope if this paper. I think it could be shorten and make it more concise.
The English language has improved significantly, but minor changes are still required.
Reviewer 3 Report
The authors timely addressed all the issues. After the revision, I have no more comments to make.
Author Response
We are pleased to know that our revisions met your expectations.
As the journal's system still requires a response from us, we are writing to formally express our gratitude for your time and expertise in reviewing our work. We are open to any additional comments as we have made some minor modifications based on the second round of comments from Reviewer 2.